# Investigation of Ultrasonic Velocity and Transmission Losses in Graphite Rods Based on Numerical Simulation and Experiment

**Mi Li [1,2] and Jianjun He [1,***

1 School of Automation Changsha, Central South University, Changsha 410083, China
2 School of Engineering and Design Changsha, Hunan Normal University, Changsha 410081, China
* Correspondence: jjhe@csu.edu.cn

**Abstract:** The depth of electrodes inserted into the charge plays a vital role in controlling the submerged arc furnace. Therefore, we used ultrasound waves reflected from the electrode tip to estimate the depth of electrodes inserted into the charge. However, graphite's ultrasonic velocity and transmission loss are the basis for establishing an ultrasonic measurement system. Thus, we expected to improve our understanding of them through numerical simulations and experimental measurements. First, we proposed an ultrasonic detection method to estimate the electrode length by embedding graphite rods in Söderberg electrodes. Then, we developed a 3D finite element model in COMSOL for wave transmission in the graphite rod. The wave transmission through 20 and 40 cm graphite rods was simulated using finite element models. The transmission loss, sound pressure, intensity, and displacement distribution of the sound wave passing through the graphite electrode were calculated. To verify the simulation calculation results, we further conducted an acoustic experiment. The results showed that transmission loss varies significantly with frequency. When the frequency was between 25 and 55 kHz, the transmission loss of the graphite rod was slight. At 47 kHz, the transmission loss was 1.837 dB/m.

**Keywords:** graphite; numerical modeling; transmission loss; ultrasonic-guided wave

## 1. Introduction

Submerged arc furnaces are critical pieces of metallurgical equipment used for ferroalloy smelting. Ferroalloy produced by submerged arc furnaces comprises more than 80% of their total production [1]. Electrical energy is delivered into the furnace through an electrode during the smelting process. The arc heat at the electrode end and the resistance heat between the material or slag to are used to melt and reduce the ore [2]. Therefore, the depth electrodes inserted into the charge directly affects a furnace's power consumption and melting temperature distribution [3]. The operator adjusts the depth of the electrode inserted into the charge by raising and lowering the electrode. However, the furnace temperature can reach up to 1350 degrees, and the electrodes are deeply buried in the charge. Therefore, measuring the depth of the electrodes into the charge is a significant challenge for many researchers.

Since the last century, researchers have made many attempts to measure the electrode length. For example, workers have estimated the length of the electrode by touching the electrode to a metal bar during smelting. This method is exceptionally unsafe and inaccurate. From the published literature, we noticed that researchers tried to estimate the electrode length by the arc current, the electrode's weight, and the magnetic field's distribution around the electric furnace. A U.S. patent attempted to estimate the electrode length by establishing a functional relationship between the electrode current and the electrode depth in the furnace [4]. However, the electrode currents of each phase are coupled with each other. Changing one electrode current will affect the electrode currents of the two different phases. Therefore, the feasibility of this method is weak.

The method using the weight of the electrode to estimate the electrode length is affected by the electrode clamping system, the addition of electrode materials, and the contact support of the charge, which makes the calculated result error too large [5]. Methods based on electric, magnetic, and temperature fields in the furnace to estimate the electrode length need to overcome the influence of electromagnetic interference, which is still in the theoretical research stage [6]. Researchers have used numerical simulation technology to analyze the relationship between the electrode length and the magnetic field radiation around the electric furnace [7,8]. However, the experimental results are not ideal. The method using electromagnetic waves to detect the electrode length requires adding a hollow waveguide to the electrode [9]. The hollow waveguide will cause the combustible furnace gas to overflow, which is fatal to safe production. Fortunately, ultrasonic-guided wave technology has been successfully applied to measure concrete anchor rods [10,11]. Ultrasonic-guided wave technology is not only used to measure the length of a bolt but also to evaluate its anchoring quality. These achievements mean that ultrasonic-guided wave technology has excellent potential in measuring consumable electrodes in submerged arc furnaces.

Therefore, we used ultrasound waves reflected from the electrode tip to estimate the electrode length. In our previous work, we analyzed the effect of high temperature on the ultrasonic velocity in graphite [12]. However, the ultrasonic transmission loss in graphite is one of the foundations for establishing an ultrasonic measurement system, because the transmission loss directly affects the distance of the sound wave transmission and the intensity of the echo. The measurement standards of material transmission loss include the pulse–echo or spectral ratio method [13,14]. However, experimental investigation of transmission loss is very time-consuming and costly. In addition, because obtaining transducers of various frequencies is challenging, only a few parameters can be studied. In contrast to experimental studies, numerical simulations allow studies with lower cost and time implications. Research results have shown that modeling-guided wave propagation using the finite element method (FEM) is effective [15]. Therefore, Seco [16] applied this method to simulate the generation and propagation of ultrasonic signals in cylindrical waveguides. Puthillath [17] simulated ultrasonic-guided wave propagation across waveguide transition. Jankauskas [18] applied the numerical simulation method to analyze the propagation of ultrasonic waveguide waves through the lap-welded plate. A study that modeled the ultrasonic cleaning technique for cylinders was conducted by Habiba [19]. Based on the simulated pressure distribution data, Habiba completed the optimal design of the sensor and achieved good application results [20]. Beata [21] modeled the propagation of ultrasonic-guided waves in anchor rods partially embedded in concrete and used simulation results to evaluate the potential of ultrasonic-guided waves for bolt debonding detection. YU [22] and Díaz [23] modeled ultrasonic propagation in concrete and acoustic attenuation in hollow concrete bricks, respectively. The simulation data are in good agreement with the experimental test data. In summary, it is desirable to analyze the propagation of ultrasound in solids by numerical simulation. Thus, we expect to improve our understanding of the ultrasonic transmission loss in graphite rods through numerical simulations and experimental measurements.

The organization of this paper is as follows. The theoretical background is given in Section 2, while Section 3 consists of experimental laboratory settings and FEM. Experimental and numerical results are discussed in Section 4, followed by conclusions in Section 5. Further work is suggested in Section 6.

The novelties and contributions of this paper are listed as follows:

(1) We proposed an ultrasonic non-destructive measurement method for consumable electrode length;
(2) We developed a FEM model which is based on the successful ultrasonic measurement results.

## 2. Theoretical Background

### 2.1. Electrode System of Submerged Arc Furnaces

Figure 1 shows the principle of the three-phase electrode system of submerged arc furnaces. The electrical energy is delivered into the furnace through the electrode. The arc heat at the electrode end and the resistance heat between the material or slag are used to melt and reduce the ore [2]. During the smelting process, the molten metal is increased by the melting of the ore, and the electrodes are consumed by corrosion. Therefore, it is necessary to raise and lower the electrode to adjust the position of the electrode ends so that the furnace maintains the proper smelting temperature [24]. However, because of the extremely high temperature in the furnace, there still needs to be suitable measurement techniques to determine the position of the electrode's working end [25]. In addition, because of mechanical stress and thermal deformation, cracks may occur in the electrode during the smelting process. If these hidden cracks are not found in time, serious accidents can easily be caused. In this paper, the method we considered was the use of ultrasonic-guided waves to detect the position of the electrode's working end and the hidden cracks in the electrode. However, the critical basis for using ultrasonic-guided wave technology is that there is a channel suitable for ultrasonic propagation in the electrode.

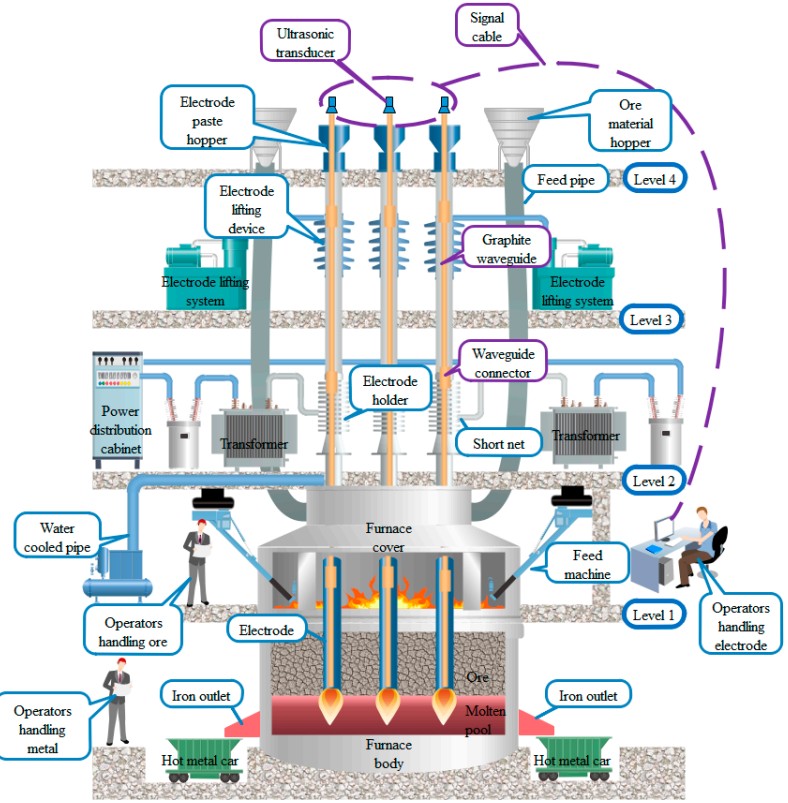

**Figure 1.** The electrode system of submerged arc furnaces with a graphite rod.

### 2.2. Ultrasonic Non-Destructive Measurement Method for Consumable Electrodes

Figure 2 shows a cross-sectional view of the consuming electrode in a medium-sized metallurgical electric furnace. The maximum electrode diameter reaches 2 m, the total length is close to 25 m, and the weight is 65 t. In the smelting process, the pre-made solid electrode material is put into the electrode casing installed on the submerged arc furnace. Then, the electrode is sintered using heat during the smelting process. Therefore, the electrode material in the electrode casing will be melted into a liquid from the original solid and then sintered into a solid. It means that from the top of the electrode to the end, the material in the electrode casing is sparse and uneven. This is detrimental to the propagation

of ultrasonic waves. Based on the abovementioned reasons, we proposed a method to form a uniform and continuous detection channel in the electrode by adding a graphite rod.

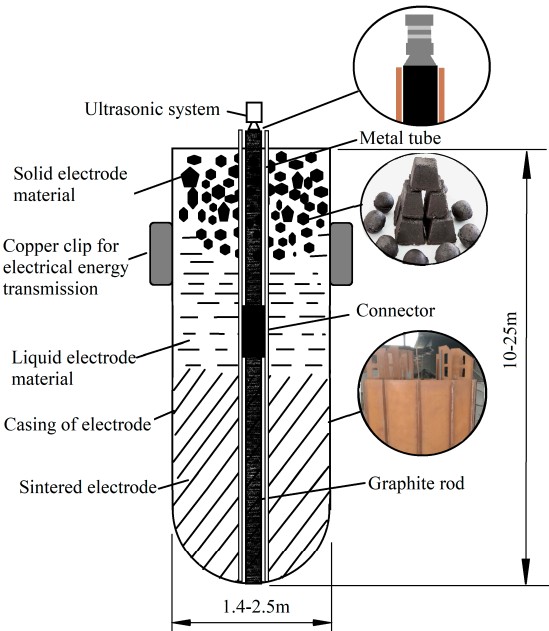

**Figure 2.** Cross-sectional view of the consuming electrode.

The measurement principle of ultrasonic-guided wave technology is as follows. The ultrasonic transducer emits the ultrasound signal on top of the electrode. The emitted ultrasound signal will propagate to the end of the electrode through the graphite rod in the electrode. Because of interface reflection, the ultrasonic transducer will receive the ultrasonic signal reflected from the electrode end. The measurement system will evaluate the parameters of the electrode based on the reflected ultrasound signal. The ultrasonic measurement system can calculate the position of the electrode working end by measuring the flight time between the transmitted signal and the echo signal. Although the shape of the electrode tip in a submerged arc furnace is similar to an ellipse [26], the graphite rod's diameter is much smaller than that of the electrode's, so we can regard the graphite rod's end face in the molten pool as a plane. When the electrode cracks, part of the ultrasound signal will be reflected in the ultrasonic transducer by the crack in advance. Therefore, the measurement system can detect hidden cracks in the electrode based on the ultrasonic signal reflected in the ultrasonic transducer in advance.

A metal tube should be added outside the graphite rod to avoid damaging the graphite rod when adding the electrode paste. The metal tube can protect the graphite rod. At the same time, adding a metal tube can form a clear interface between the metal tube and the graphite rod, which is conducive to reducing the diffusion and radiation of sound waves to the electrode in the graphite rod. In addition, the inner diameter of the metal tube should be greater than the outer diameter of the graphite rod so that the cylindrical surface of the graphite rod can be considered as a free boundary.

When the ultrasonic signal propagates in the solid, some factors will attenuate the signal amplitude, such as the geometric dispersion caused by the columnar structure, the beam spread, the material's porosity, etc. [27]. The detection accuracy will be greatly reduced if the echo amplitude is too low. The electrode length in the medium-sized electric furnace is usually more than 10 m, which makes the non-destructive monitoring for consumable electrodes different from the general ultrasonic flaw detection. Obtaining sufficient echo amplitude is a significant challenge. However, a larger detection distance can be obtained by optimizing the ultrasonic transducer in the non-destructive detection of bolts [28]. However, the propagation characteristics of ultrasonic signals in graphite are different from those of structural steel. The most significant difference is that the ultrasonic velocity

in graphite increases instead of decreasing with increasing temperature [12]. Therefore, analyzing the acoustic characteristics of ultrasonic waves propagating in graphite rods is necessary. The results can be used to evaluate whether ultrasonic-guided wave technology can be used for the non-destructive monitoring of consumable electrodes.

## 3. Experimental Setup and FEM Theory

To evaluate the potential of using ultrasonic guided waves to monitor consumable electrodes in submerged arc furnaces, we need to know more details regarding the ultrasonic propagation in high-density graphite rods, such as stress and strain distribution and ultrasonic transmission loss. In this study, we used finite element analysis combined with experimental measurements to study the propagation of ultrasonic waves in graphite rods. First, we analyzed the propagation velocity and transmission loss of ultrasonic waves with different frequencies in the graphite rod through experimental measurements. Since obtaining transducers with different frequencies and powers is challenging, experimental measurements can only analyze a few parameters. Therefore, we used COMSOL to calculate the sound field distribution in the graphite rod. Additionally, the validity of the numerical analysis was verified by comparing the numerical simulation results with the experimental measurement results.

### 3.1. Experimental Set-Up

As shown in Figure 3, the ultrasonic measurement system used in this paper was composed of a Langevin transducer, signal generator, power amplifier, impedance matcher, filter circuit, data acquisition card, and PC. The ultrasonic measurement experiment adopts the discontinuous measurement method. In the experiment, we used 10 cycles of sine signal to excite the transducer. The amplitude of the excitation signal was 500 V and the pulse width was 0.04 ms.

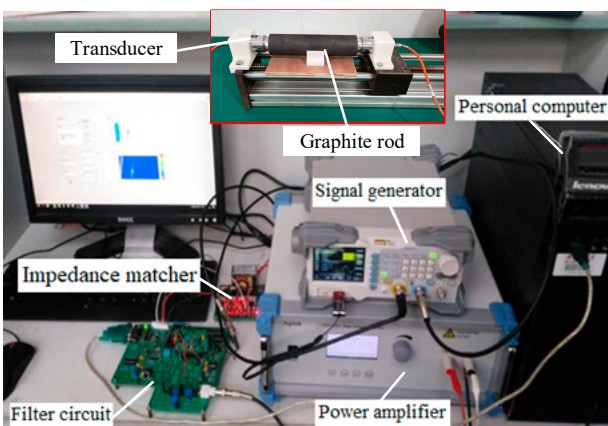

**Figure 3.** Ultrasonic measuring system.

At room temperature, we placed the graphite electrode sample between two Langevin transducers (as shown in Figure 4). At the beginning of the measurement, the signal generator output a sinusoidal signal with a predetermined frequency and then amplified the signal through a power amplifier, which used an impedance matcher to drive the transmitting transducer to work. At the same time, the data acquisition card was triggered by the external clock to record the signal received from the receiving transducer and save it to the PC. The number of received signal sampling points was 512, and the sampling period was 0.4 microseconds. To reduce the interference noise in the received signal, we used a low-pass filter circuit to filter the output of the receiving transducer.

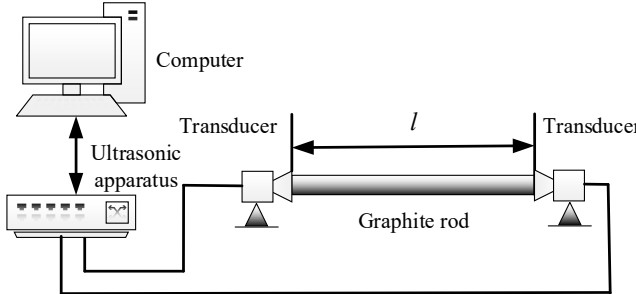

**Figure 4.** Illustration of the ultrasonic measuring system.

　　To select the frequency of the transducer in the experiment, it is necessary to know the propagation velocity of each guided wave mode of the graphite rod. Therefore, we obtained the group velocity curve of the graphite rod with a diameter of 50 mm by using the semi-analytical finite element method [29], as shown in Figure 5. Please refer to Appendix A for the theory of guided wave mode in rod.

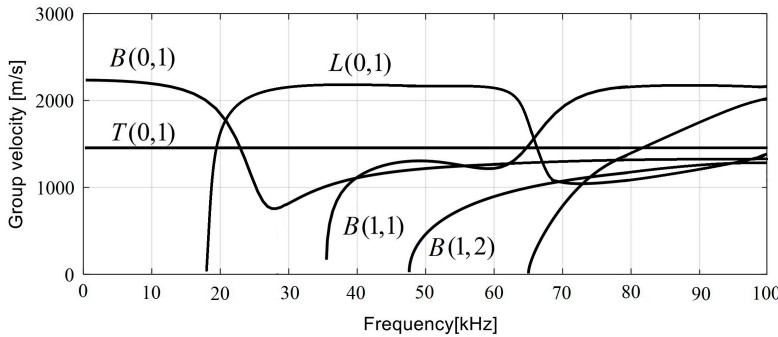

**Figure 5.** Group velocity dispersion curve of the graphite rod.

　　The following can be seen from Figure 5:

(1)　Different modes had different degrees of dispersion. The same mode had different degrees of dispersion in different frequency ranges;
(2)　The phase velocity of the $T$ (0, 1) mode was frequency-independent, which indicates that the $T$ (0, 1) mode was a non-dispersive mode;
(3)　Near the cutoff frequency of the $L$ (0, 1) longitudinal mode, there was a frequency range with a small degree of dispersion, and the $L$ (0, 1) mode had the largest group velocity in this frequency range. In summary, the group velocity of the $L$ (0, 1) mode was larger than that of other modes in the range of its low-frequency dispersion frequency. This means that the $L$ (0, 1) mode signal will reach the transducer first.

　　The group velocity value of $L$ (0, 1) mode was the largest in the frequency range of 25–60 kHz. Therefore, we selected two endpoints of this frequency range for experimental measurement. In addition, we added 47 kHz as the test point near the midpoint of 25–60 kHz. When the frequency was 100 kHz, the group velocity value of $B$ (1, 1) mode was the largest, so we also took 100 kHz as the test point.

　　The equipment list of the ultrasound system is as follows:

　　The model of the signal generator is DG1022 Z;

　　The model of the power amplifier is ATG-2021. The maximum output voltage is 170 Vpp, the maximum output power is 42.5 W, the input impedance is 50 Ω/5 KΩ, and the output impedance is 5 Ω/50 Ω;

　　The model of the data acquisition card is PCI-1714. The highest sampling frequency is 30 MS/s, and the resolution is 12 bits;

　　We used Langevin transducers with frequencies of 25 kHz, 47 kHz, 60 kHz, and 100 kHz. The beam angles of the transducers are 94°, 58°, 53°, and 42° respectively;



We used a low-pass filter circuit with a cutoff frequency of 250 kHz.

### 3.2. Graphite Rod Sample

In this paper, the sample was cut from graphite electrodes used in metallurgy. The raw materials of graphite electrodes mainly include solid carbon raw materials and binders. Solid carbon raw materials usually contain 50% anthracite, 30% petroleum coke, and 20% crushed graphite. The binder is coal pitch and coal tar. The manufacturing process of graphite electrodes can be divided into five steps: raw material calcination, crushing, mixing, pressure forming, and baking [30]. Raw material calcination is performed by putting solid carbon raw materials into a furnace isolated from the air for calcination. Crushing is carried out to process the calcined electrode material into fine powder. The mixing step comprises mixing the electrode raw materials and the binder to produce a paste-like electrode raw material. Pressure forming involves putting the mixed electrode raw materials into extrusion molding equipment and processing them into a predetermined shape. Baking involves putting the formed electrode into a baking furnace with a protective medium for heating and baking according to the baking process. According to the manufacturing process, the graphite rod is a carbon material with a disordered orientation texture. Therefore, the graphite rod is homogeneous and isotropic.

The theoretical density of the graphite electrode sample is 1870 kg/m$^3$, the Young's modulus is 8.5 GPa, and the void volume fraction is 17%. The sample length is 200 mm and 400 mm, and the diameter is 50 mm. To ensure better coupling between the transducer and the test piece, the two ends of the test piece were turned flat by a lathe. We placed the sample between the transmitting and receiving transducer to measure the ultrasonic velocity.

### 3.3. Layout of the FEM Model

To understand the propagation of sound waves in the graphite rod, we established a steady-state finite element model in the frequency domain in COMSOL. We used this model to analyze the transmission characteristics of ultrasonic waves with different frequencies between 25 kHz and 100 kHz. The model was established based on the sound transmission loss measurement standard [31]. The finite element model consists of a source room, a receiving room, and a graphite rod, as shown in Figure 6. The left is the source room, and the right is the receiving room. Both the source room and the receiving room are filled with air. We set the surroundings of the sound source room and the receiving room as a perfect matching layer (PML) to simulate the infinite air domain. The thickness of the perfect matching layer was set as the maximum wavelength. We used pressure acoustics, solid mechanics, and acoustic–structure coupling boundaries in the model. The sound source room and the receiving room were set as pressure acoustics. The graphite rod was set as the solid mechanic. The graphite rod material was set to conform to linear elastic behavior. The attenuation of ultrasonic waves in graphite was simulated by adding an isotropic loss factor in the material properties. We used the cantilever beam resonance method [32] to measure the isotropic loss factor of graphite, which is 0.012. The boundary between the fluid and solid domains was set as the acoustic–structure coupling boundary, which simulates the coupling between pressure acoustics and solid mechanics.

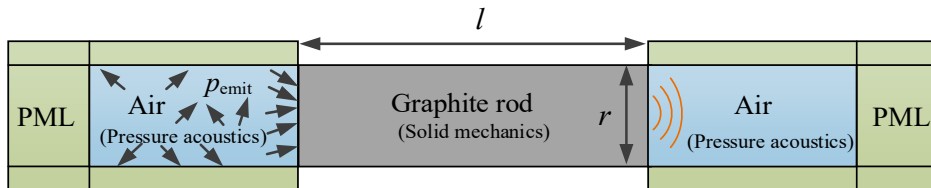

**Figure 6.** Model setup of a graphite electrode.

For pressure acoustics, we used the wave equation shown in Equation (1). We selected $1 \times 10^{-6}$ Pa as the reference pressure of the sound pressure level. We set the model to atmospheric pressure and temperature and the transient pressure acoustic model to linear

elasticity. We set the velocity of sound in the air as 346 m/s and the density of the air as 1.205 kg/m$^3$.

$$\nabla \cdot \left( -\frac{1}{\rho_c} (\nabla p_t - q_d) \right) - \frac{\omega^2 p_t}{c_c^2 \rho_c} = Q_m \tag{1}$$

where, $\rho_c$ is the density of the air, $p_t$ is the total pressure, $q_d$ is the dipole domain source, and $Q_m$ is the unipolar domain source.

To simulate the excitation of sound sources with different frequencies between 25 kHz and 100 kHz, we defined the sound field of the sound source room as the sum of $N$ uncorrelated plane waves moving in random directions. Using $N$ waves in different directions to excite sound waves can avoid the necessity of modeling the transducer, which can reduce the calculation amount of the model. If the transducer is used, it will not be easy to define the model because it has different structural dimensions because of different frequencies.

The pressure field of the sound source room was determined by the following formula:

$$
\begin{aligned}
p_{emit} &= \frac{1}{\sqrt{N}} \sum_{n=1}^{N} \exp(-i(k_{n,x}x + k_{n,y}y + k_{n,z}z)) \exp(i\Phi_n) \\
k_{n,x} &= \cos(\theta_n) \\
k_{n,y} &= \sin(\theta_n)\cos(\varphi_n) \\
k_{n,z} &= \sin(\theta_n)\sin(\varphi_n)
\end{aligned}
\tag{2}
$$

where, $\theta_n$, $\varphi_n$, and $\Phi_n$ are the direction angles of the plane wave, and they are independent random numbers. Each plane wave has a different direction angle. The $1/\sqrt{N}$ term is used to ensure that any $N$ has a constant strength.

Therefore, the incident sound intensity in the $z$-axis direction of the sound source room can be obtained as follows:

$$
\begin{aligned}
I_{in} &= \frac{1}{4}\mathrm{Re}(p_{emit}, v_{emit}) \\
v_{emit} &= \frac{-1}{i\omega\rho}\frac{\partial p_{emit}}{\partial z}
\end{aligned}
\tag{3}
$$

where, $p_{emit}$ is the pressure field of the sound source chamber, $v_{emit}$ is the velocity in the $z$-axis direction, $\rho$ is the density of air, and $\mathrm{Re}(a, b)$ is the dot product of complex numbers $a$ and $b$.

The sound intensity in the $z$-axis direction of the receiving room can be defined as follows:

$$I_{re} = \frac{1}{2}\mathrm{Re}(p_t, i\omega w) \tag{4}$$

where, $w$ is the displacement of the structure in the $z$-axis direction, $p_t$ is the total sound pressure at the receiving end, and $\mathrm{Re}(a, b)$ is the dot product of complex numbers $a$ and $b$.

The sound transmission loss of the medium is defined as the ratio of the sound intensity in the $z$-axis direction of the sound source room and the receiving room. According to Equations (3) and (4), the sound transmission loss of the graphite rod can be obtained as follows:

$$a(\omega) = 20\lg[I_{in}/I_{re}]/l \tag{5}$$

where, $I_{in}$ is the incident sound intensity in the $z$-axis direction of the sound source chamber, as defined in Equation (3). $I_{re}$ is the sound intensity in the $z$-axis direction of the receiving room, as defined in Equation (4). $l$ is the length of the graphite rod.

A suitable mesh is critical for analyzing the propagation characteristics of steady-state frequency domain acoustic waves. The wave equation's solution depends on the mesh's appropriate size. To solve the wave equation to produce an accurate solution, about 10 or 12 nodes are required for each wavelength. It means that each wavelength includes at least five second-order meshes. Therefore, the maximum mesh size can be set as $l_{\max} = c_0/(5f_{\max})$, where $c_0$ is the velocity of sound and $f_{\max}$ is the maximum frequency of the wave. When the highest frequency is used to limit the maximum mesh size, the mesh used for low-frequency analysis will be too dense. Although dense meshes help

improve calculation accuracy, excessively dense meshes will consume too many computing resources. Therefore, the size of the smallest mesh should be reasonably limited. We set the minimum mesh size as $l_{min} = c_0/(6f_{max})$. As shown in Figure 7, it is the mesh setting of the model. We set the blue surface in Figure 7 as the "Mapped" mesh type. According to the sound velocity in the air and the maximum frequency $f_{max}$, the number of cells in the direction of the red line is 22. The remaining models were set to the "Swept" mesh type. The number of cells in the length direction of the source and receiving room was 67. The number of cells in the length direction of the graphite rod was 178.

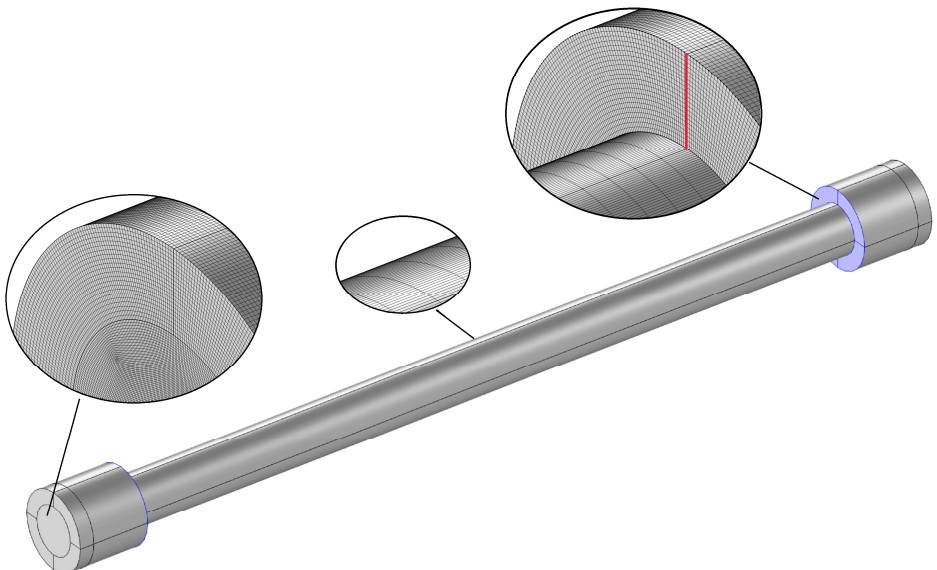

**Figure 7.** Mesh settings for the model.

## 4. Results and Discussions

### 4.1. Experimental Measurement Results

Using the measuring device shown in Figure 3, the velocity and transmission loss of ultrasonic waves passing through the graphite rod at four different frequencies, 25 kHz, 47 kHz, 60 kHz, and 100 kHz, respectively, were measured. The sample diameter was 50 mm, and the length was 200 mm and 400 mm, respectively. The two ends of the sample were processed flat and placed between the transmitting transducer and the receiving transducer. We applied butter between the transducer and the sample to realize the coupling between the transducer and the sample. We used Equation (6) to calculate the ultrasonic propagation velocity in the graphite electrode according to the first wave arrival time in the received signals of the two samples. Using the ratio of length and time difference to calculate the ultrasonic propagation velocity can reduce the influence of measuring equipment error on measuring results.

$$c = \frac{l_2 - l_1}{t_2 - t_1} \tag{6}$$

where, $l_1$ = 200 mm, $t_1$ is the first wave arrival time of the 200 mm sample, $l_2$ = 400 mm, and $t_2$ is the first wave arrival time of the 400 mm sample.

Figure 8 shows the time and frequency response of the received signal in a 400 mm sample. The ultrasonic velocity in the graphite rod was 2275 m/s when the working frequency of the transducer was 25 kHz, 47 kHz, 60 kHz, and 100 kHz, respectively; the ultrasonic velocity in the graphite rod changed by only 2%, which is within the measurement error.

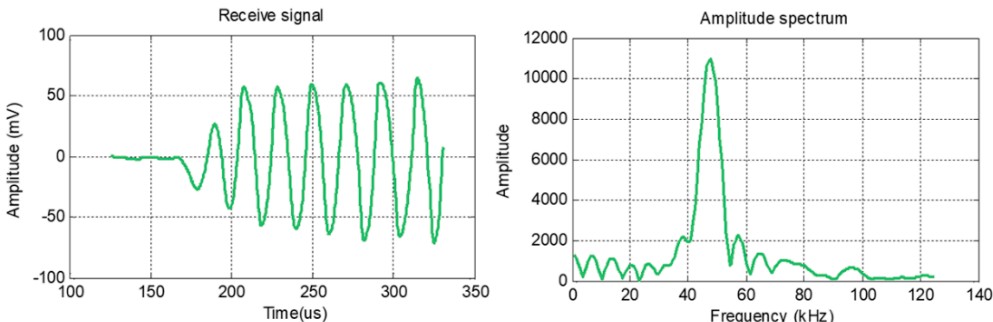

**Figure 8.** The time and frequency response of the received signal in a 400 mm sample.

When ultrasonic waves pass through the medium, the lattice structure will absorb part of the energy and reduce the wave amplitude. This attenuation is caused by the type of material, small holes, and small cracks. For long-distance measurements, attenuation is a major problem. Therefore, it is necessary to study the transmission loss of ultrasonic waves in graphite rods. This paper used Equation (7) to calculate the transmission loss. Figure 9 shows the time and frequency response of the received signal with a frequency of 47 kHz in different long test samples.

$$a(\omega) = \frac{20\lg A_1(\omega) - 20\lg A_2(\omega)}{l_2 - l_1} \tag{7}$$

where, $A_1(\omega)$ and $A_2(\omega)$ are the amplitude spectrum of the received signals in two samples with different lengths, and $l_1$ and $l_2$ are the length of the samples, respectively.

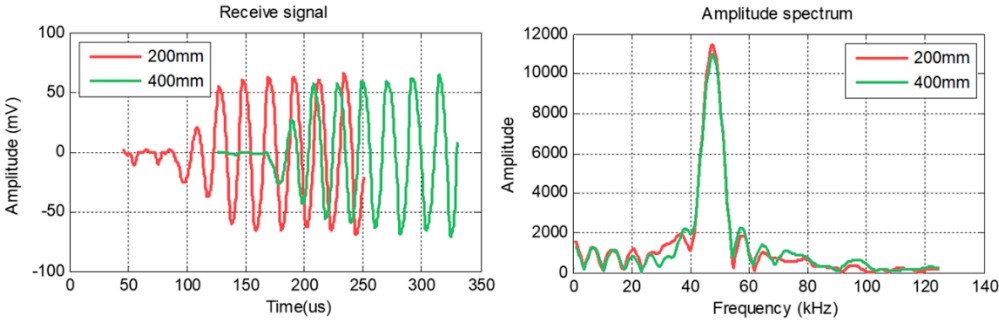

**Figure 9.** The time and frequency response of the received signal in two samples.

*4.2. Numerical Simulation Results*

Figure 10 shows the calculated and measured group velocity. From Figure 10, we know that when the excitation frequencies were 25 kHz, 47 kHz, and 60 kHz, and the measured value is the group velocity of $L$ (0, 1) mode. When the excitation frequency is 100 kHz, the measured value is the group velocity of $B$ (1, 1) mode. The measured value has the same trend as the theoretical calculated value.

As shown in Figure 11, the transmission loss of ultrasonic waves with different frequencies was calculated by Equation (5). The blue curve in Figure 11 results from numerical simulation, and the green curve is the result of experimental measurement. Although the simulated transmission loss (shown by the blue curve in Figure 11) exhibited fluctuations, it does not mean that the solutions did not converge. This is because we defined the sound field of the sound source room as the sum of $N$ uncorrelated plane waves moving in random directions. As shown in Figure 12, there was an expected significant fluctuation at low frequencies where the wavelength was comparable to the rod diameter, because at low frequencies, sound waves cannot diffuse effectively. This means that the sound field will not diffuse under the measurement conditions under these low frequencies.

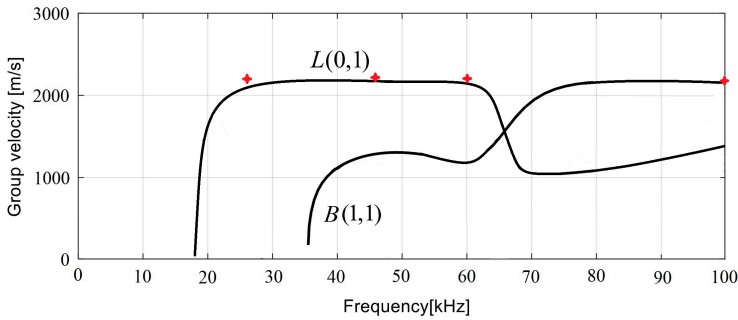

**Figure 10.** The calculated and measured group velocity.

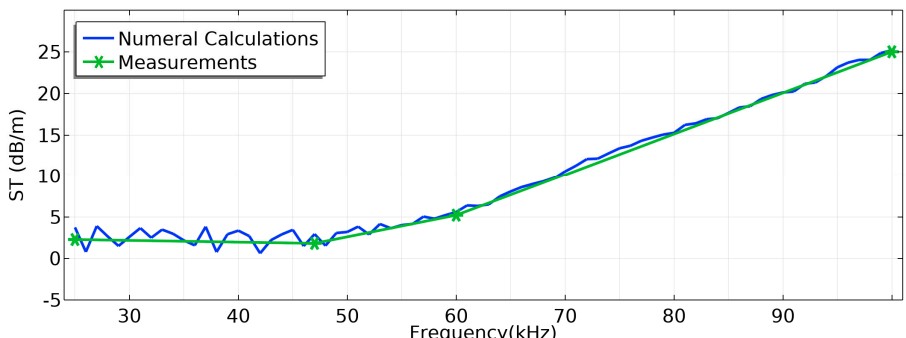

**Figure 11.** Transmission loss of different frequency ultrasonic waves.

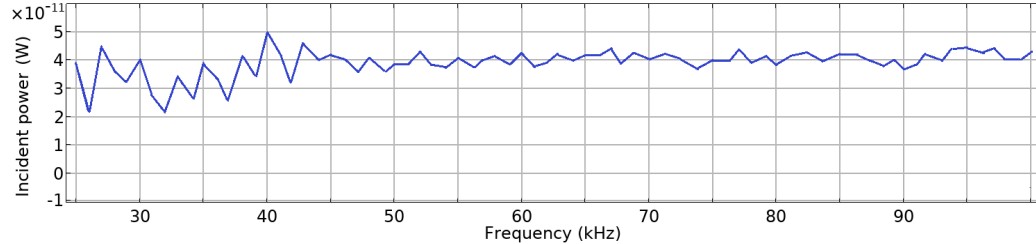

**Figure 12.** Incident power.

As shown in Figure 11, we know that the numerical simulation and experimental measurement results have the same change trend. Comparing the numerical simulation with the experimental data, the two agree. It shows that the numerical simulation method is reliable for calculating the ultrasonic transmission loss in graphite rods.

From Figure 11, we know that when the frequency is low, the transmission loss of ultrasonic waves through the graphite rod is slight. When the ultrasonic wave frequency is 47 kHz, the transmission loss is only 1.837 dB/m. This is beneficial for reducing the attenuation of the sound wave amplitude and increasing the detection distance. As the ultrasonic wave frequency increases, the transmission loss will increase exponentially with the frequency. A higher ultrasonic wave frequency will cause the ultrasonic wave to decay quickly.

In the frequency range of 25–55 kHz, the transmission loss changed relatively smoothly, and the transmission loss almost did not change with the increased acoustic frequency. In this frequency range, the transmission loss was insensitive to the ultrasonic wave frequency. The frequency in this range can be used as the transducer's resonant frequency.

The incident intensity and transmitted intensity in the graphite rod surface are shown in Figure 13. The ultrasonic wave frequencies were 25 kHz, 47 kHz, 60 kHz, and 100 kHz. It can be seen from Figure 13 that the incident intensity distribution was only related to the randomness and the number of plane waves $N$ in Equation (2). The transmitted intensity was determined by the sound pressure and displacement at the receiving end of

the graphite rod. The displacement is related to the transferable modes in the graphite rod. The sound pressure and displacement at the receiving end of the graphite rod are shown in Figure 14.

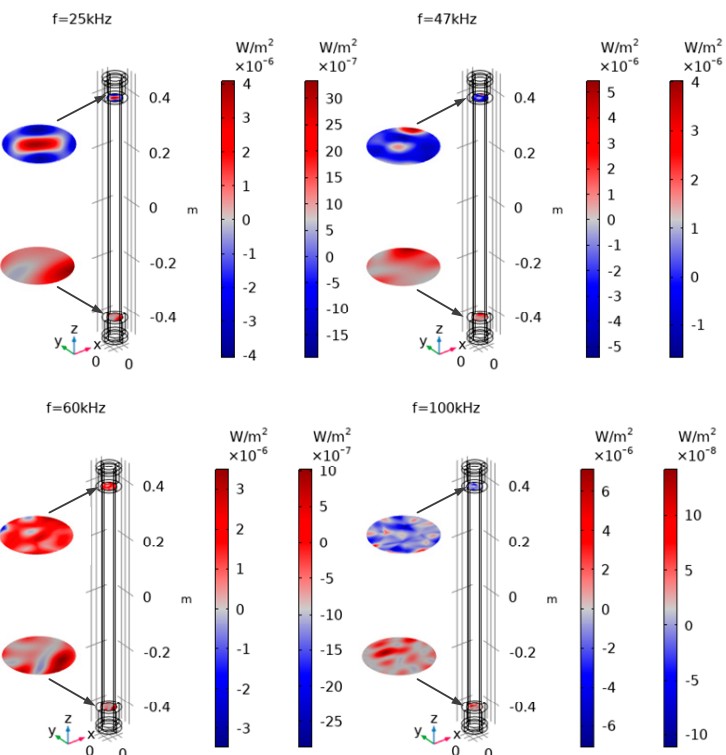

**Figure 13.** The incident intensity and transmitted intensity in the graphite rod surface.

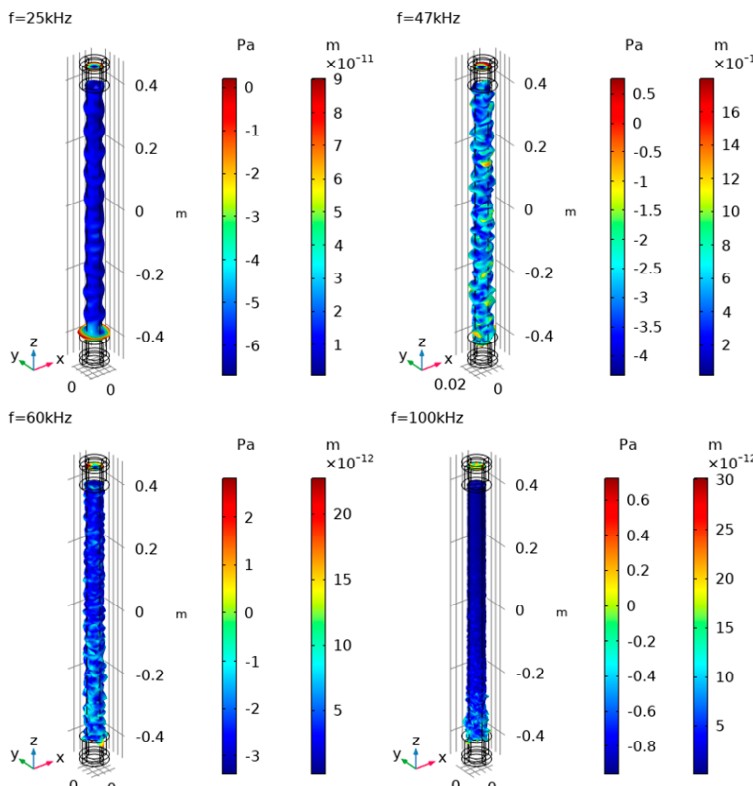

**Figure 14.** The displacement and sound pressure at the receiving end.

## 5. Conclusions

This work discussed the ultrasonic velocity and transmission loss in graphite rods. Based on solid elastic mechanics, the differential equation of motion of longitudinal waves propagating in a viscoelastic rod was derived. Additionally, the derived differential equations of motion were used to make COMSOL numerical models.

Using the experimental ultrasonic measurement device, we successfully measured graphite rods' ultrasonic velocity and transmission loss at different frequencies. The ultrasonic velocity in the graphite rod was 2275 m/s at the four selected frequency points. The transmission loss varied significantly with frequency emission. The graphite rod had a smaller transmission loss when the ultrasonic wave frequency was between 25 and 55 kHz. At 47 kHz, the transmission loss was only 1.837 dB/m. This means that when the ultrasonic wave propagated 10 m, it still had 12% intensity.

The COMSOL model is based on a double silencing terminal, which includes a sound source room and a receiving room filled with air. The influence of ultrasonic reflection on the calculation of sound field intensity was eliminated. The results showed that the transmission loss calculated by the COMSOL model was in good agreement with the experimental measurement results. This provides a practical and effective calculation method for estimating the sound transmission loss in the rod. Comparing the transmitted intensity drawn in COMSOL with the displacement of the structure, it was determined that the displacement of the structure controls the transmitted intensity.

## 6. Further Work

The COMSOL model and experimental measurements neglected the mechanical connection between graphite rods, which may affect the transmission of ultrasonic waves. As electrodes are consumed during smelting, new graphite rods must be added to the electrodes using suitable connectors. Therefore, further investigation is a must to understand the effects of different forms of connectors on ultrasonic transmission.

The measurement experiment and simulation results of the transmission loss showed that the electrode non-destructive ultrasonic-guided wave monitoring has application potential. However, optimization is the next step to improve the detection distance and distance resolution. The proposed COMSOL model can be used to determine the best transducer power, frequency, excitation signal type, and other input parameters.

Further investigation on the effects of coupling between the transducer contact surface and the graphite rod is needed to measure the changes caused by different contact forms such as acoustic coupling gel or dry contact. In addition, by identifying the critical factors in ensuring maximum echo intensity, the equipment used to install transducers could be optimized for the end user's convenience.

**Author Contributions:** Conceptualization, J.H. and M.L.; methodology, M.L.; software, M.L.; validation, J.H. and M.L.; formal analysis, M.L.; writing—original draft preparation, M.L.; writing—review and editing, M.L.; visualization, M.L.; supervision, J.H.; project administration, J.H.; funding acquisition, J.H. All authors have read and agreed to the published version of the manuscript.

**Funding:** This work was sponsored by the National Natural Science Foundation of China, No. 61873282, and the Scientific research Project of the Education Department of Hunan Province No. 22 C0007.

**Institutional Review Board Statement:** Not applicable.

**Informed Consent Statement:** Not applicable.

**Data Availability Statement:** Not applicable.

**Conflicts of Interest:** The authors declare no conflict of interest.

## Appendix A. The Propagation of Elastic Waves in the Rod

To study the dispersive propagation of elastic waves in an infinite rod, Kolskyt provided the motion equation of the element in a rod based on the Navier–Stokes equation in cylindrical coordinates [33].

$$(\lambda + 2\mu)\frac{\partial \phi}{\partial r} - \frac{2\mu}{r}\frac{\partial \omega_z}{\partial \theta} + 2\mu\frac{\partial \omega_\theta}{\partial z} = \rho\frac{\partial^2 u_r}{\partial t^2} \tag{A1}$$

$$(\lambda + 2\mu)\frac{1}{r}\frac{\partial \phi}{\partial \theta} - 2\mu\frac{\partial \omega_r}{\partial z} + 2\mu\frac{\partial \omega_z}{\partial r} = \rho\frac{\partial^2 u_\theta}{\partial t^2} \tag{A2}$$

$$(\lambda + 2\mu)\frac{\partial \phi}{\partial z} - \frac{2\mu}{r}\frac{\partial(r\omega_\theta)}{\partial r} + \frac{2\mu}{r}\frac{\partial \omega_r}{\partial \theta} = \rho\frac{\partial^2 u_z}{\partial t^2} \tag{A3}$$

where, $\mu$ is Poisson's ratio, and $u_r$, $u_\theta$, and $u_z$ are the radial, circumferential, and axial displacement components of the rod, respectively; $\phi$ is the volume invariant in cylindrical coordinates, and $\omega_r$, $\omega_\theta$, and $\omega_z$ are the three components of the rotation vector.

According to the motion equation given by Graff, the vibration displacement in the rod conforms to the law of simple harmonic vibration.

$$u_r = U(r)\cos n\theta e^{i(kz-\omega t)} \tag{A4}$$

$$u_\theta = V(r)\sin n\theta e^{i(kz-\omega t)} \tag{A5}$$

$$u_z = W(r)\cos n\theta e^{i(kz-\omega t)} \tag{A6}$$

where, $U(r)$, $V(r)$, and $W(r)$ are the radial, circumferential, and axial displacement amplitude of the rod respectively, $\theta$ is the circumference angle, $\omega$ is the angular frequency; $k$ is wave number, $n = 0, 1, 2 \cdots$ is the order of the circumference direction.

When $n = 0$, Equations (A4)–(A6) are the axisymmetric propagation of longitudinal waves in the rod. Figure A1 shows a schematic diagram of the longitudinal axisymmetric mode.

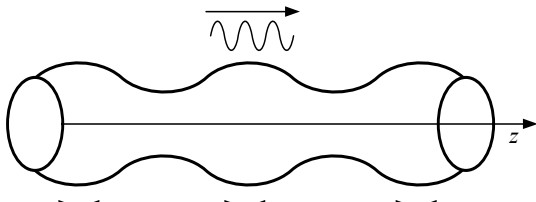

**Figure A1.** Longitudinal axisymmetric mode.

The frequency equation of the longitudinal propagation mode is as follows [34]:

$$\frac{2\alpha}{a}\left(\beta^2 + k^2\right)J_1(\alpha a)J_1(\beta a) - \left(\beta^2 - k^2\right)^2 J_0(\alpha a)J_1(\beta a) - 4k^2\alpha\beta J_1(\alpha a)J_0(\beta a) = 0 \tag{A7}$$

where, $\alpha^2 = \frac{\omega^2}{c_L^2} - k^2$, $\beta^2 = \frac{\omega^2}{c_T^2} - k^2$, and $a$ are the radius of the rod, $k$ is the wave number, $c_L$ is the longitudinal wave velocity, $c_T$ is the transverse wave velocity, $J_0()$ is the zero-order Bessel function, and $J_1()$ is a first-order Bessel function.

When the radial and axial displacement components of the rod are zero ($u_r = 0$, $u_z = 0$), Equations (A4)–(A6) are the propagation of torsional waves in the rod. Figure A2 shows a schematic diagram of the torsional mode.

The frequency equation of the torsional mode is as follows [35]:

$$(\beta a)J_0(\beta a) - 2J_1(\beta a) = 0 \tag{A8}$$

where, $\alpha^2 = \frac{\omega^2}{c_L^2} - k^2$, $\beta^2 = \frac{\omega^2}{c_T^2} - k^2$, and $a$ are the radius of the rod, $k$ is the wave number, $c_L$ is the longitudinal wave velocity, $c_T$ is the transverse wave velocity, $J_0()$ is the zero-order Bessel function, and $J_1()$ is a first-order Bessel function.

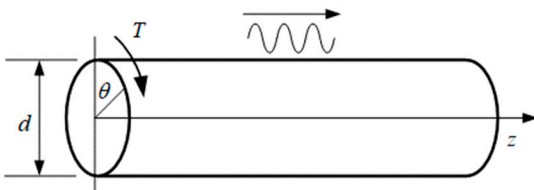

**Figure A2.** Torsional mode.

From the frequency equation of torsional mode, it is known that torsional mode is the non-dispersive mode.

$n = 1$ denotes the most important bending mode of the rod. Figure A3 shows the bending mode of the rod. For the frequency equation of bending mode, Pao [36,37] and Kumar [38] conducted in-depth research.

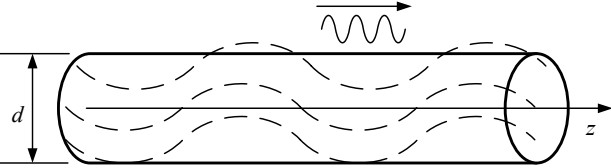

**Figure A3.** Bending mode.

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
