# Peer review of "Investigation of Ultrasonic Velocity and Transmission Losses in Graphite Rods Based on Numerical Simulation and Experiment"

_applsci, doi:10.3390/app13053329_

Round 1

Reviewer 1 Report

The study presented in this paper is very interesting. In fact, it proposes a solution for the electrode length measurment in furnaces which represent a big problem in metallurgy industry. however, several points should be corrected in order to level the paper up to publication standards:

1) there is some phrases that have to be reformulated as in page 1 lines 33 to 35.

2) in subsection 3.3: the parameters choices in the model should be argumented 

3) In page 8, need of a mesh sensitivity study and a need of a mesh image with arguments on the mesh structure choice. 

4)  source chamber or source room

5) the fluctuation in numerical results could be due to mesh?? 

6) i can not see the significance of the results presented in figure 10. 

7) Graphite not graphic line327 page 9

8) in the model description the authors have declared that the two rooms were filled with air but in the conclusion, it is mentionned that the receiving and source rooms are filled with water. Which one is correct? 

Author Response

Dear Editor,

Our paper, applsci-2242975, "Investigation of ultrasonic velocity and transmission losses in graphite rod based on numerical simulation and experiment" has been revised according to each reviewer’s comments, and the itemized replies to their comments and the changes made in the revised paper are attached.

The article has been checked carefully and thoroughly, many parts have been adjusted to better express the idea, and some writing mistakes have been corrected too. For your convenience, all the changes have been highlighted in red color and all language changes have been highlighted in blue in the revised paper.

Many thanks for your valuable processing and the reviewer’s comments so that we can further improve our paper.

We look forward to hearing from you.

Sincerely,

The authors

Name:  Jianjun He

Tel: +86 13203106175

E-mail:[email protected]

Our replies to each reviewer’s comments are appended below.

Reviewer 2 Report

The work contain numerous shortcomings and should be reviewed by authors carefully once again. Some details are indicated in the attached file.

The laboratory set as well as numerical model deviate significantly form real physical bacground depicted in Figure 1. It is necessary to discuss the diferences. Are the rods in the molten pool flat?

Author Response

(The authors gave the same response as above.)

Round 2

Reviewer 1 Report

I recommend the acceptance of the paper in the present form.  The authors have correctly responded to the review comments.